# Mechanical forces couple bone matrix mineralization with inhibition of angiogenesis to limit adolescent bone growth

Maria Dzamukova [1,2 ✉], Tobias M. Brunner[1,2], Jadwiga Miotla-Zarebska[3], Frederik Heinrich [4], Laura Brylka [5], Mir-Farzin Mashreghi [4,6], Anjali Kusumbe [7], Ralf Kühn [8], Thorsten Schinke [5], Tonia L. Vincent[3] & Max Löhning [1,2 ✉]

Bone growth requires a specialised, highly angiogenic blood vessel subtype, so-called type H vessels, which pave the way for osteoblasts surrounding these vessels. At the end of adolescence, type H vessels differentiate into quiescent type L endothelium lacking the capacity to promote bone growth. Until now, the signals that switch off type H vessel identity and thus limit adolescent bone growth have remained ill defined. Here we show that mechanical forces, associated with increased body weight at the end of adolescence, trigger the mechanoreceptor PIEZO1 and thereby mediate enhanced production of the kinase FAM20C in osteoblasts. FAM20C, the major kinase of the secreted phosphoproteome, phosphorylates dentin matrix protein 1, previously identified as a key factor in bone mineralization. Thereupon, dentin matrix protein 1 is secreted from osteoblasts in a burst-like manner. Extracellular dentin matrix protein 1 inhibits vascular endothelial growth factor signalling by preventing phosphorylation of vascular endothelial growth factor receptor 2. Hence, secreted dentin matrix protein 1 transforms type H vessels into type L to limit bone growth activity and enhance bone mineralization. The discovered mechanism may suggest new options for the treatment of diseases characterised by aberrant activity of bone and vessels such as osteoarthritis, osteoporosis and osteosarcoma.

[1] Pitzer Laboratory of Osteoarthritis Research, German Rheumatism Research Centre (DRFZ), a Leibniz Institute, Berlin, Germany. [2] Experimental Immunology and Osteoarthritis Research, Department of Rheumatology and Clinical Immunology, Charité – Universitätsmedizin Berlin, corporate member of Freie Universität Berlin and Humboldt-Universität zu Berlin, Berlin, Germany. [3] Centre for Osteoarthritis Pathogenesis Versus Arthritis, Kennedy Institute of Rheumatology, University of Oxford, Oxford, UK. [4] Therapeutic Gene Regulation, Regine von Ramin Lab Molecular Rheumatology, German Rheumatism Research Centre (DRFZ), a Leibniz Institute, Berlin, Germany. [5] Department of Osteology and Biomechanics, University Medical Center Hamburg-Eppendorf, Hamburg, Germany. [6] BIH Center for Regenerative Therapies (BCRT), Charité – Universitätsmedizin Berlin, Berlin, Germany. [7] Tissue and Tumour Microenvironments Group, University of Oxford, Oxford, UK. [8] Max-Delbrück-Center for Molecular Medicine in the Helmholtz Association (MDC), Berlin, Germany. ✉email: maria.dzamukova@drfz.de; max.loehning@charite.de

Proper skeleton development requires a fine-tuned crosstalk between different types of cells and tissues present in bone[1]. Juvenile bones harbour highly angiogenic type H vessels, which are closely associated with and coordinate the differentiation of bone growth-mediating osteoblasts. In contrast, quiescent type L vessels are present in the adult bone[2–4]. Type H vessels are located exclusively at the sites of active bone growth, namely the ossification front (OF) and periosteum, and they are organised in columnar structure[2]. Notch signalling, Hypoxia-inducible factor 1-alpha (HIF1α) activity, blood flow speed and slit guidance ligand 3 (SLIT3) were shown to support type H vessel formation[3,5,6]. However, the mechanism that directs the switch from type H to type L vessels resulting in bone growth cessation has remained unknown. A recent study reported that endothelial cells in bone require integrins for their proper functioning and maintenance, which highlights extracellular matrix (ECM) proteins as important factors for endothelium activity[7]. Yet, little is known about further ECM proteins controlling endothelial cells in bone. Therefore, we analysed changes in the composition of ECM proteins throughout bone development.

In this work we describe the molecular mechanism that links mechanical forces and angiogenesis in bone via mechanically triggered accumulation of the extracellular matrix protein dentin matrix protein 1 (DMP1) and its regulation of vascular subtypes.

## Results

### DMP1 is a potential candidate controlling endothelial cell fate in bone.
To dissect age-related changes in vascular and perivascular microenvironments during and after active bone growth, we performed laser microdissection (LMD) of single capillary with associated surrounding cells in the OF of juvenile (4-week-old) and adult (12-week-old) mice. We developed a protocol to cut single capillary from undecalcified and unfixed bones of endothelial cell-specific GFP-reporter mice with a high RNA preservation rate to subsequently perform next-generation sequencing (NGS) (Supplementary Fig. 1a, b). Principal component analysis showed that the cells from the juvenile OF were very similar between different mice, whereas with ageing, greater heterogeneity was apparent (Supplementary Fig. 1c). To identify candidates from the list of differentially expressed genes with a potential of regulating type H vasculature, we focused on ECM proteins (Fig. 1a, b and Supplementary Fig. 1d). DMP1 appeared to be a promising candidate because of its reported angiogenesis-modulating activity in a retina tumour model[8].

DMP1 is known as an essential player in matrix mineralisation[9,10]. Our detailed analysis of DMP1 protein localisation throughout postnatal bone development showed that until 5 weeks of age, the protein was mainly localised at the base of the zone of type H vessels in the OF (Fig. 1c, d). Unexpectedly, at the age of 5.5–6 weeks, a burst in DMP1 secretion into ECM occurred primarily in the central top part of the OF also reaching the growth plate (GP) (Fig. 1c, e–g and Supplementary Fig. 2a). We performed quantifications of protein staining intensities within central and peripheral regions of interest in the OF (Fig. 1f, g and Supplementary Fig. 2b). During late postnatal bone maturation with increasing age, the DMP1-occupied area spread from the centre towards the periphery of the OF. Finally, by the age of 12 weeks, the whole metaphysis together with the GP were interspersed with DMP1 (Fig. 1c, f, g). This behaviour was seen irrespective of sex.

To identify the main cellular source of DMP1 in the OF-GP area at different ages, we established a method for RNAScope in bone. Co-hybridisation was performed with probes for *Dmp1* together with *Runx2* for early osteoblasts, osterix (*Osx* or *Sp7*) for mature osteoblasts and sclerostin (*Sost*) for osteocytes

(Supplementary Fig. 2c). Sclerostin expression was not found in the OF but only in the cortical bone. At all analysed stages in the OF-GP region, *Dmp1* mRNA was only detected in the OF area but never in the chondrocytes of the GP, including time points directly before (5 weeks) and during the burst in DMP1 protein secretion (5.5 weeks) (Fig. 1h and Supplementary Fig. 2f). *Dmp1* mRNA was mainly found in *Osx*+ cells and to a lesser extent in *Runx2*+ cells at all examined time points, which identifies osteoblasts as the main producer of DMP1 in the metaphysis (Supplementary Fig. 2d, e). The number and intensity of *Dmp1*-expressing cells peaked at 4–5 weeks (Fig. 1i, j and Supplementary Fig. 2g). Yet, there was no difference at the level of *Dmp1* gene expression between the central and peripheral parts of the OF, indicating that the observed differences in DMP1 amounts between these regions emerged rather at the protein level (cf. Fig. 1c, f, g). This observation goes in line with previous reports that the control of DMP1 availability and function mainly takes place at the posttranslational rather than transcriptional level[11–13] explaining the discrepancy in the timing of mRNA and protein peaks (Fig. 1c, f, g, i, j). Together, these data demonstrate that the burst in DMP1 secretion from osteoblasts in the OF centre at the age of 5.5–6 weeks is controlled at the protein level and DMP1 likely diffuses from the OF to the GP (Fig. 1k).

### DMP1 transforms type H vessels into type L.
In addition to secreting osteogenic factors[3], tip cells of type H vessels secrete matrix metalloproteinase 9 (MMP9) that digests cartilage matrix and allows vessel bulges to invade GP cartilage, thereby facilitating bone growth[4]. We noticed that DMP1 accumulation in the OF centre at 5.5–6 weeks correlated with a decrease in endomucin (EMCN) intensity and reduced amounts of MMP9 at the OF-GP border. This was associated with the disappearance of bulges and columnar structure of the vessels (Fig. 2a, b and Supplementary Fig. 3). Hence, all main features of type H vessels were downregulated. Importantly, at 5.5–6 weeks, these changes occurred only in the central part of the OF.

To study whether DMP1 directly regulates bone angiogenesis and growth, we generated mice with a constitutive and an inducible deletion of *Dmp1* (Supplementary Fig. 4). Constitutive *Dmp1*-deficient mice were smaller with shortened and widened long bones (Supplementary Fig. 4c–f) as had been described before[9,14], and they featured a disorganized OF and GP (Supplementary Fig. 5a). The blood vessels in the OF were dilated and had very high EMCN and MMP9 intensity, even in adulthood, though the columnar structure was lost already in juvenile bones (Supplementary Fig. 5c, d). This finding suggests that, presumably via its matrix mineralisation activity, DMP1 provides a scaffold for columns of type H vessels. The phenotype of *Dmp1*+/− mice was identical to *Dmp1*+/+ mice, which indicates that one *Dmp1* allele generates sufficient DMP1 protein (Supplementary Figs. 4c, d and 5b). This result is consistent with human genetic studies where it was shown that mutations in the *DMP1* gene cause autosomal recessive hypophosphatemia[15]. OSX+ and RUNX2+ osteoblasts were present in the OF to a similar extent as in the wild-type group indicating that DMP1 is not required for osteoblast differentiation and maturation[16] (Supplementary Fig. 5a, b). Notably, the GP of *Dmp1*−/− mice exhibited an expanded pre-hypertrophic and hypertrophic zone (RUNX2+ and SOX9+) (Supplementary Fig. 5a). We assume that this phenotype is caused by the lack of proper vessel invasion due to the loss of vessel architecture.

To test whether DMP1 induces the switch from type H to type L vessels, we generated a *Dmp1*-floxed mouse line and crossed it with *Col1-CreERT2* mice to ablate *Dmp1* in an inducible manner selectively in osteoblasts (*Dmp1*iΔOB) upon tamoxifen injection

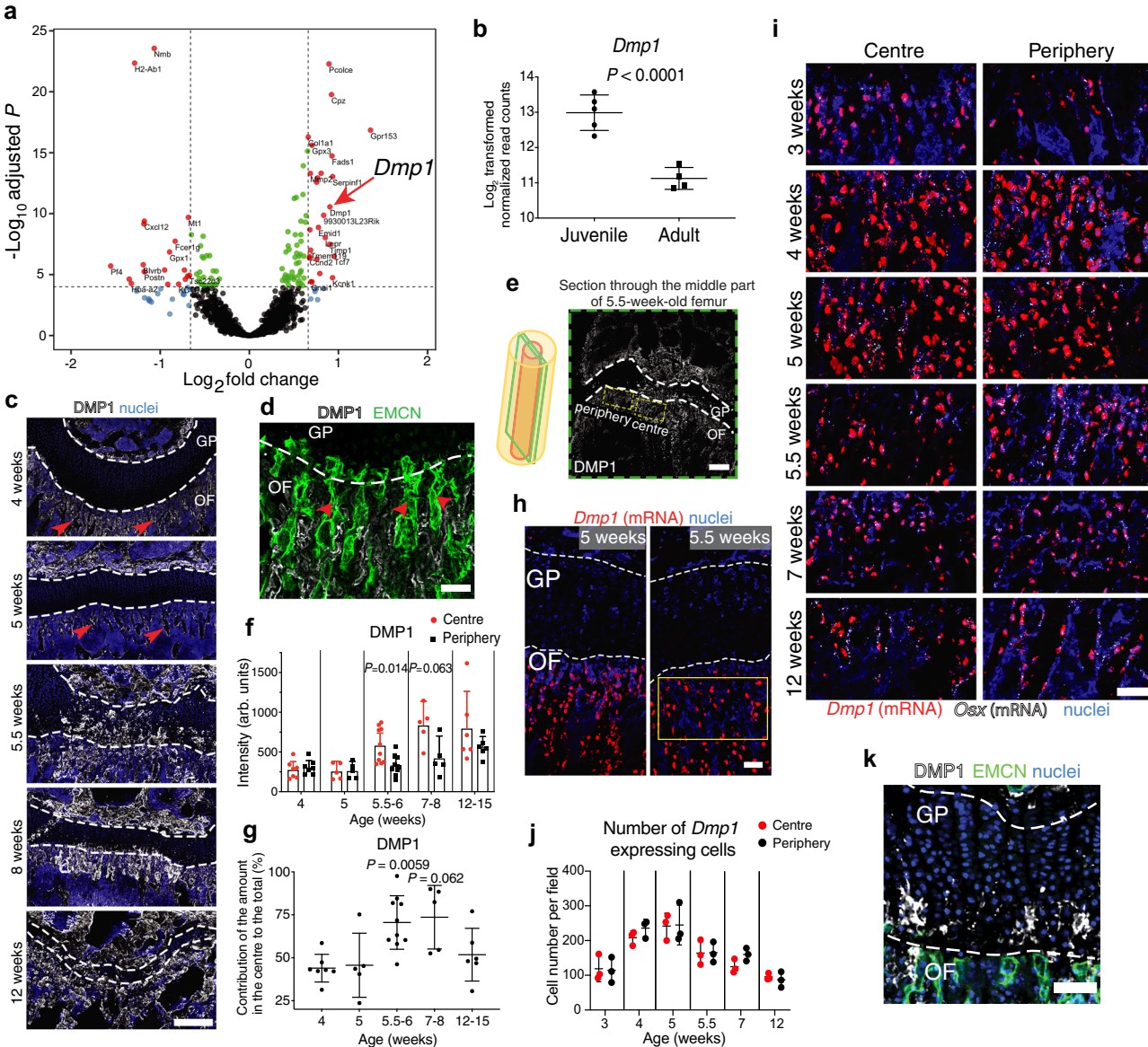

**Fig. 1 Burst in DMP1 secretion from osteoblasts into ECM at the OF-GP border during periadolescence. a** Volcano plot of significantly differentially expressed genes in juvenile (4 weeks, $n = 5$) versus adult (12 weeks, $n = 4$) capillary with surrounding cells from the ossification front (OF) (mean expression value >500, red: $Log_2$ FC > 0.66, $P < 10^{-5}$). **b**, $Log_2$ transformed number of reads mapped to the *Dmp1* gene. **c** DMP1 immunostaining in the OF throughout postnatal bone development. Red arrows point at DMP1 at the OF base. Scale bar 250 μm. **d** DMP1 and endomucin (EMCN) immunostaining of the OF from 5-week-old femur. Red arrows point at type H vessels. Scale bar 50 μm. **e** Schematic depiction of the femur with a green plane going through the centre of the bone, where DMP1 secretion is observed. Example of DMP1 immunostaining of a section taken from the middle part of the femur, showing the definition of the central and peripheral regions used for further automated quantification. Scale bar 250 μm. **f** Quantification of DMP1 staining intensity in the centre and periphery of the OF throughout bone development. Mean ± SD. **g** Quantification of the contribution of the central and peripheral part to the total DMP1 amount. Mean ± SD. **f**, **g** at 4 weeks $n = 7$, 5 weeks $n = 5$, 5.5–6 weeks $n = 10$, 8 weeks $n = 5$, 12–15 weeks $n = 6$. Pooled data from five independent experiments. $P$ values show the significance between the intensity in the centre versus periphery for every mouse (paired analysis). Two-tailed Wilcoxon matched-pairs signed-rank test. **h** *Dmp1* mRNA (RNAScope) of the central regions of OF and GP from 5- and 5.5-week-old femurs. Lack of *Dmp1* expression in chondrocytes of the GP; yellow frame indicates the region shown in **i**. Scale bar 50 μm. **i** RNAScope images of *Dmp1*, Osterix (*Osx*), and *Runx2* (only depicted in Supplementary Fig. 2c). Scale bar 50 μm. **j** Number of *Dmp1*-expressing cells in the central and peripheral region of the OF based on RNAScope images. $n = 3$ per age. Pooled data from two independent experiments. Mean ± SD. **k** DMP1 and EMCN immunostaining of 5.5-week-old femur. Infiltration of DMP1 from the OF into the GP. Scale bar 50 μm. **b**, **f**, **g**, **j** Source data are provided in Source Data file.

(Supplementary Fig. 4g–i). Deletion of *Dmp1* was performed at P28-32 to allow normal formation of the OF together with the columnar structure of blood vessels (Fig. 2c). Tamoxifen treatment led to ~75% reduction of *Dmp1* mRNA expression by week 6 (P42) (Supplementary Fig. 6a, b) and reduced the amount of DMP1 protein in the OF centre by ~70% (Fig. 2d and Supplementary Fig. 6c). The body weight was not affected

(Supplementary Fig. 6d). Yet, long bones of *Dmp1*[iΔOB] mice featured an enlarged metaphysis with overall increased EMCN and MMP9 intensities (Fig. 2e, f and Supplementary Fig. 6e). In line with the femur phenotype of constitutive *Dmp1*-deficient mice (cf. Supplementary Fig. 4d, f), also *Dmp1*[iΔOB] femurs were slightly shorter than control femurs by the age of 6 weeks, presumably reflecting the key role of DMP1 in matrix

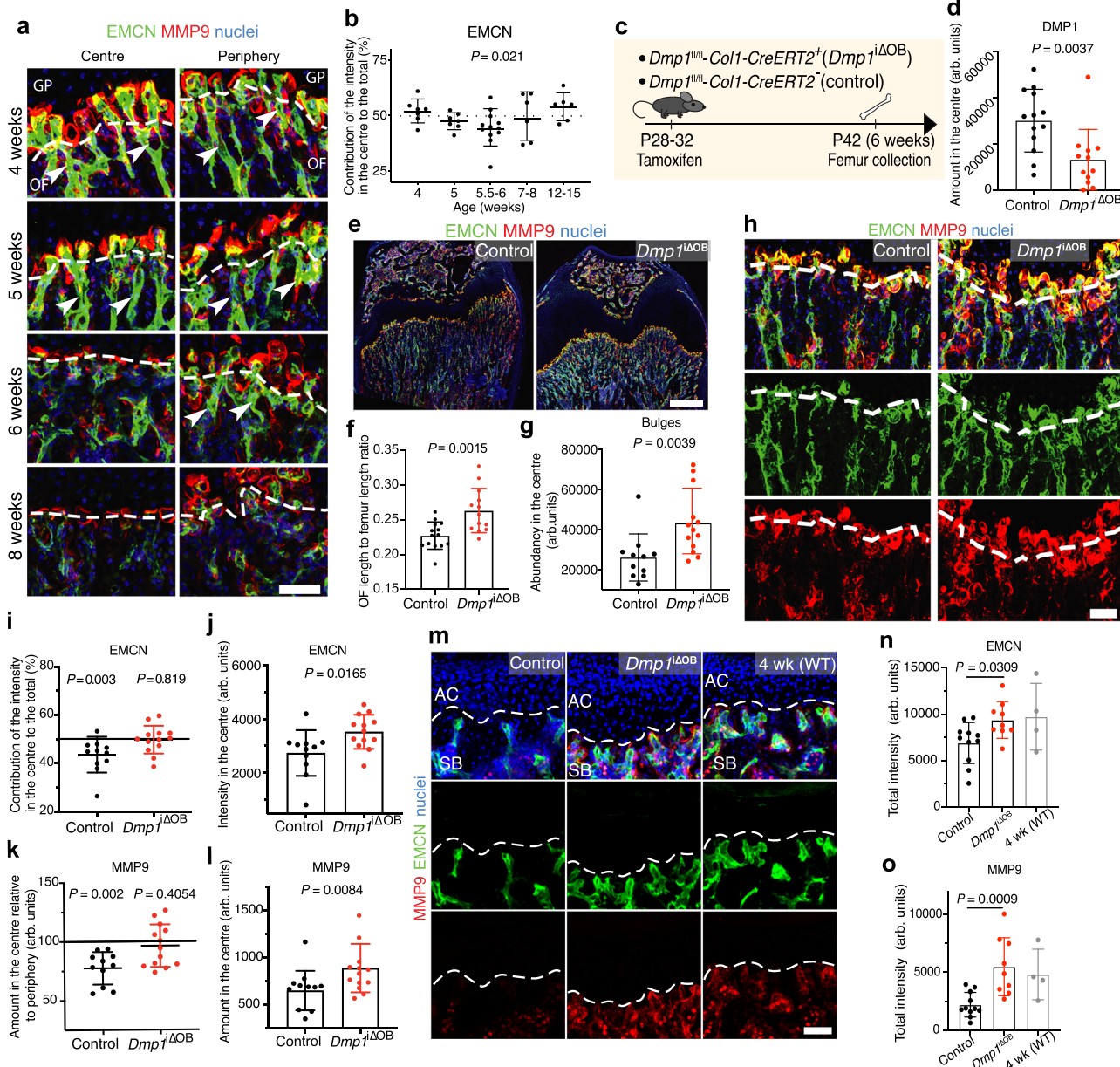

**Fig. 2 DMP1 transforms active type H vessels into quiescent type L in the OF and in subchondral bone. a** EMCN and MMP9 immunostaining of femur. Dashed line shows the OF-GP border. White arrows point at type H vessels. Scale bar 50 μm. **b** Quantification of the contribution of the central and peripheral parts to the total EMCN intensity. At 4- and 5-weeks $n = 7$, 5.5–6 weeks $n = 12$, 8 weeks $n = 6$, 12–15 weeks $n = 6$. Pooled data from six independent experiments. $P$ value shows the significance between the intensity in the centre versus periphery for every mouse (two-tailed paired $t$-test). Mean ± SD. **c** Experimental scheme to induce *Dmp1* deletion by Tamoxifen injection. **d** Quantification of DMP1 amount in the OF centre of control and osteoblast-specific inducible *Dmp1*-deleted (*Dmp1*$^{iΔOB}$) mice. Two-tailed Mann–Whitney test. **e** EMCN and MMP9 immunostaining of 6-week-old control and *Dmp1*$^{iΔOB}$ femurs. Scale bar 500 μm. **f** Quantification of ossification front length to femur length ratio in control and *Dmp1*$^{iΔOB}$ mice. **d, f** Control $n = 14$, *Dmp1*$^{iΔOB}$ $n = 13$. **g** Quantification of bulges by measuring the total EMCN intensity in the central GP in control and *Dmp1*$^{iΔOB}$ mice. Two-tailed Mann–Whitney test. **h** EMCN and MMP9 immunostaining of femur in 6-week-old control and *Dmp1*$^{iΔOB}$ mice. Dashed line shows the OF-GP border. Scale bar 50 μm. **i** Quantification of the central to peripheral intensity ratio of EMCN in control and *Dmp1*$^{iΔOB}$ OF. **j** Quantification of the total central EMCN intensity in control and *Dmp1*$^{iΔOB}$ group. **k** Quantification of MMP9 at the OF-GP border in control and *Dmp1*$^{iΔOB}$ femurs. **i–k** Two-tailed paired $t$-test. **l** Quantification of the total central MMP9 amount at the OF-GP border in control and *Dmp1*$^{iΔOB}$ group. Two-tailed Mann–Whitney test. **g–l** Pooled data from three independent experiments. Control $n = 11$, *Dmp1*$^{iΔOB}$ $n = 13$. Age: 6 weeks. Mean ± SD. **m** EMCN and MMP9 immunostaining of 6-week-old control, *Dmp1*$^{iΔOB}$ and 4-week-old WT subchondral bone (SB) with articular cartilage (AC). Dashed line indicates the SB-AC border. Scale bar 50 μm. **n** Quantification of the total EMCN intensity at the SB-AC border. **o** Quantification of the total MMP9 intensity around the vessels at the SB-AC border. **m–o** Pooled data from two independent experiments. Control $n = 11$, *Dmp1*$^{iΔOB}$ $n = 9$, 4-week-old WT $n = 4$. Mean ± SD. **b, d, f, g, j–l, n, o** Source data are provided in Source Data file.

mineralisation and bone architecture (Supplementary Fig. 6f). Nevertheless, detailed analysis showed that the deletion of *Dmp1* in osteoblasts arrested the age-related maturation of the OF. Specifically, type H vessels together with bulges were preserved in the central part of the OF in *Dmp1*[iΔOB] mice (Fig. 2g, h). Moreover, the difference in EMCN and MMP9 intensity between the central and peripheral parts of the OF was absent in *Dmp1*[iΔOB] femurs (Fig. 2i, k). The overall EMCN intensity and MMP9 amount in the OF centre were higher in *Dmp1*[iΔOB] samples compared with the control group (Fig. 2j, l). As expected, von Kossa staining showed reduced mineralisation of the OF in *Dmp1*[iΔOB] femurs, confirming the crucial importance of DMP1 for bone calcification (Supplementary Fig. 6g, h, j).

In line with the phenotype of *Dmp1*[iΔOB] mice at the OF, analysis of subchondral bone revealed that *Dmp1* deletion at 4 weeks led to preserved density of EMCN[+] vessels and high amounts of MMP9 around the vessel tips underneath the articular cartilage by week 6 (Fig. 2m-o). Control mice featured significantly reduced vessel activity in subchondral bone, whereas *Dmp1*[iΔOB] samples were very similar to 4-week-old wild-type subchondral bone. Thus, deletion of *Dmp1* at 4 weeks prevented the development-associated switch from active to quiescent vessel fate not only at the OF but also in subchondral bone. The mineralisation of subchondral bone was also reduced in *Dmp1*[iΔOB] femurs (Supplementary Fig. 6g, i, k). Hence, we cannot formally exclude that the reduced mineralisation contributes to some extent to the observed phenotype in *Dmp1*[iΔOB] bones. Nevertheless, our results suggest that DMP1 is involved in the inhibition of angiogenesis in long bones and in the transformation of active type H into quiescent type L vessels, which results in reduced EMCN and MMP9 expression and loss of vessel bulges.

**DMP1 inhibits VEGF signalling in the ossification front in vivo and in endothelial cells in vitro.** Vascular endothelial growth factor (VEGF) signalling is crucial for angiogenesis and bone growth[17–19]. The proliferation of the chondrocytes in the proliferative layer of the GP allows the bone to elongate, whereas the hypertrophic chondrocytes in the bottom layer express high amounts of VEGFA, which drives the invasion of type H vessels into the GP, thus leading to the progression of the ossification front[17,18]. To test whether DMP1 has VEGF-inhibiting properties in bone, we studied phosphorylation of the main VEGF receptor VEGFR2 (pVEGFR2) along with DMP1 accumulation in ECM. Local DMP1 amounts negatively correlated with the amount of pVEGFR2 (Fig. 3a and Supplementary Fig. 7). The VEGFR2 phosphorylation level in 4- to 5-week-old mice was equal across the OF. However, at 5.5-6 weeks we observed a marked reduction in the amount of pVEGFR2 in the central part of the OF in wild-type mice. This difference was largely lost in *Dmp1*[iΔOB] mice, and the overall intensity of pVEGFR2 in the OF centre was higher (Fig. 3b–d). Of note, the total level of VEGFR2 protein was not affected by *Dmp1* deletion (Fig. 3e, g).

Previous studies showed that VEGFR2 activation induces *Flt4* (VEGFR3) expression in endothelial cells[20,21]. VEGFR3 is usually highly upregulated in endothelial tip cells, where it is required for vessel sprouting[7,20,22]. Bulges in the GP of *Dmp1*[iΔOB] mice stained strongly with VEGFR3 compared with remaining bulges of control mice (Fig. 3f, h). This result suggests that osteoblast-released DMP1 downregulates VEGFR3 in endothelial tip cells most likely by interference with VEGFR2 signalling, thus leading to the loss of vessel bulges. Further support came from the finding that treatment of the capillary endothelial cell line bEnd.3 with DMP1 abrogated the VEGF-induced phosphorylation of VEGFR2 and the expression of *Fos* (*c-fos*), a target gene of

VEGF signalling[23] (Fig. 3i–k). In summary, these data demonstrate that in the OF in vivo and in endothelial cells in vitro, DMP1 inhibits VEGF signalling, which is a known driver of angiogenesis and osteogenesis[19]. Taken together with previously published results[8], we suggest that the anti-angiogenic effect of DMP1 likely is not restricted to type H vessels in bone but it rather appears to act as a general inhibitor of angiogenesis at those sites where DMP1 is present.

**DMP1 secretion correlates positively with body weight and coincides spatiotemporally with FAM20C upregulation.** Hormones, especially sex hormones, are usually considered important regulators of bone growth and metabolism[24–26]. Mice achieve puberty by approximately week 6, which coincides in time with the burst in DMP1 secretion. We noticed that the extent of DMP1 release at 5.5-6 weeks varied considerably between individual mice (cf. Fig. 1f). We also noted that in mice with lower body weight than average, the burst in DMP1 secretion had not yet taken place at 5.5–6 weeks. Inversely, littermates heavier than average featured already very advanced DMP1 release at this time point (Fig. 4a). Indeed, we found a positive correlation between body weight and the extent of DMP1 secretion (Fig. 4b). DMP1 is a highly phosphorylated protein and a substrate of FAM20C kinase that exclusively mediates the phosphorylation and subsequent secretion of DMP1 into ECM in vivo[11,27,28]. Notably, in 6-week-old mice, we found more FAM20C, along with higher DMP1 accumulation, in the OF centre in comparison to the periphery, which was not the case in 4-week-old mice (Fig. 4c–e). In line with FAM20C acting upstream of DMP1, inducible *Dmp1* deletion did not impair FAM20C upregulation at the age of 6 weeks (Supplementary Fig. 8a, b). These findings suggest that mechanical loading through increasing body weight or/and muscle contractions, directly or indirectly, controls DMP1 secretion along with FAM20C upregulation.

**Mechanical loading enhances FAM20C kinase production and DMP1 secretion.** To investigate whether FAM20C production was enhanced by mechanical loading, we cultured live 400-500-μm-thick slices of 4-week-old femurs and centrifuged them twice daily on 5 consecutive days (Fig. 5a). Loading via centrifugation caused an upregulation of intracellular FAM20C protein and enhanced DMP1 secretion (Fig. 5b, c and Supplementary Fig. 8c, d). The mechanically induced FAM20C upregulation apparently occurred hormone-independently in the cultured femur slices in vitro.

To study if body weight- and/or muscle force-associated mechanical loading controls FAM20C production also in vivo, we unloaded one hind limb in the mouse by cutting the sciatic and femoral nerves at the age of 4.5 weeks (Fig. 5d). This results in paralysis of the affected hind limb, hence releasing it from loading through muscle contraction as well as body weight with all weight going through the contralateral (normal) side, whereas both limbs were exposed to the same hormonal environment. Unloading by double neurectomy prevented FAM20C upregulation and DMP1 secretion at 5.5–6 weeks in almost all of the operated limbs, whilst preserving normal DMP1 release on the contralateral side (Fig. 5e, f and Supplementary Fig. 8e, f). Thus, mechanical loading enhances local FAM20C production and DMP1 secretion.

**Mechanical forces trigger PIEZO1 to enhance FAM20C kinase production and DMP1 secretion.** In our transcriptome analysis, we found four mechanoreceptors that osteoblasts could potentially express in the OF during late postnatal development: *Trpv4*, *Kcnk2*, *Piezo1*, *Piezo2* (Supplementary Fig. 8g). Recently

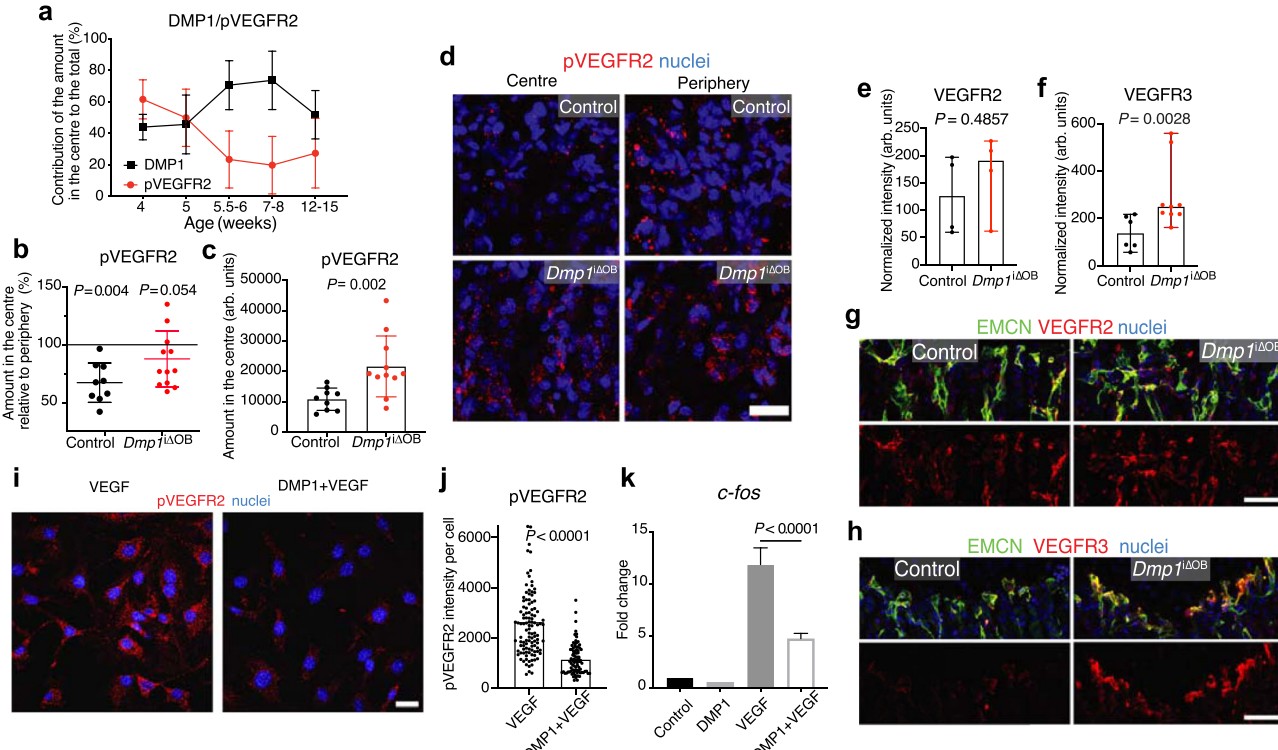

**Fig. 3 Extracellular DMP1 inhibits VEGF signalling in the OF in vivo and in endothelial cells in vitro. a** Quantification of the centre to periphery ratio of phosphorylated VEGFR2 (pVEGFR2) and DMP1 amount at different ages. At 4 and 5 weeks $n = 7$, 5.5-6 weeks $n = 6$, 8 weeks $n = 5$, 12–15 weeks $n = 4$. Pooled data from three independent experiments. Mean ± SD. **b** Quantification of pVEGFR2 amount in the centre relative to periphery in control and $Dmp1^{i\Delta OB}$ OF. Two-tailed Wilcoxon matched-pairs signed-rank test. **c**, Quantification of the total central pVEGFR2 amount in control and $Dmp1^{i\Delta OB}$ group. Two-tailed unpaired $t$-test. **b, c** Pooled data from three independent experiments. Control $n = 9$, $Dmp1^{i\Delta OB}$ $n = 11$. Age: 6 weeks. Mean ± SD. **d** pVEGFR2 immunostaining showing no decrease of pVEGFR2 in the OF centre of $Dmp1^{i\Delta OB}$ femurs in contrast to control femurs at 6 weeks of age. Scale bar 25 μm. **e** Quantification of VEGFR2 intensity in EMCN+ endothelial cells in the OF of control and $Dmp1^{i\Delta OB}$ mice. Control $n = 4$, $Dmp1^{i\Delta OB}$ $n = 4$. Pooled data from two independent experiments. Two-tailed Mann–Whitney test. Median ± range. **f** Quantification of VEGFR3 intensity in the OF of control and $Dmp1^{i\Delta OB}$ mice. Control $n = 6$, $Dmp1^{i\Delta OB}$ $n = 9$. Pooled data from three independent experiments. Two-tailed Mann–Whitney test. Median ± range. **g, h** VEGFR2 and EMCN (**g**) and VEGFR3 and EMCN (**h**) immunostaining of the OF centre in 6-week-old control and $Dmp1^{i\Delta OB}$ mice. Scale bar 50 μm. **i** pVEGFR2 immunostaining of bEnd.3 cells, which were either treated with VEGF only (left) or pre-treated with DMP1 (40 nM) for 3 h before adding VEGF (50 ng/ml). Scale bar 25 μm. **j** Quantification of the pVEGFR2 intensity per cell of bEnd.3 cells, which were either treated with VEGF only or pre-treated with DMP1 before adding VEGF. Two-tailed Mann–Whitney test. Mean. **k** qPCR of *c-fos* expression, one of the target genes of VEGF signalling, in bEnd.3 cells, which were treated either with DMP1 only, or with VEGF only, or first with DMP1 and then with VEGF. Tukey's multiple comparisons test. Mean + SD. **i–k** Representative plots of 2 independent experiments. **a–c, e, f, j, k** Source data are provided in Source Data file.

published studies show that in osteoblasts, mainly PIEZO1 and to a lesser extent PIEZO2 are key mechanosensors mediating anabolic effects of loading on bone[29–33]. Thus, we tested whether PIEZO1 or PIEZO2 control *Fam20c* upregulation in osteoblasts by analysis of bones of 6-7-week-old mice with *Piezo1* or *Piezo2* deficiency in *Dmp1*-expressing cells (*Piezo1^{ΔDmp1}* or *Piezo2^{ΔDmp1}*). FAM20C upregulation and the resulting burst in DMP1 secretion were lost exclusively in *Piezo1^{ΔDmp1}* bones but normal in *Piezo2^{ΔDmp1}* femurs (Fig. 6a–c, d–f and Supplementary Fig. 8h, i). Osteoblast maturation and abundance were not affected by *Piezo1* deletion both at 4 and 7 weeks, as determined by the staining intensity and frequency of OSX+ cells in the OF (Supplementary Fig. 8j–n). 4-week-old *Piezo1^{ΔDmp1}* bones featured similar levels of FAM20C and DMP1 as control bones indicating that PIEZO1 mediates the loading-induced upregulation of FAM20C around periadolescence, whereas basal FAM20C expression is PIEZO1 independent (Fig. 6g–i). In addition, we compared the phenotype of *Piezo1^{ΔDmp1}* bones to control ones using von Kossa/van Gieson staining followed by histomorphometry at 7 weeks of age. Both assessments did not show significant differences between the groups at this age (Supplementary Fig. 8o, p). These data suggest that the enhanced

FAM20C upregulation and DMP1 secretion in control bones versus *Piezo1^{ΔDmp1}* bones is likely to be mediated directly by PIEZO1 and not by differences in bone maturation between the two groups. To provide additional evidence and validate the involvement of PIEZO1 in the proposed mechanism, we exposed femur slices from 4-week-old *Piezo1^{ΔDmp1}* and Cre-negative (WT) littermate mice to loading via low-speed centrifugation as described (cf. Fig. 5a). Mechanical loading did not lead to upregulation of FAM20C or enhanced DMP1 release in *Piezo1^{ΔDmp1}* femur slices, while it did so in Cre-negative (WT) samples (Fig. 6j and Supplementary Fig. 8q). To conclude, mechanical forces trigger PIEZO1 to enhance FAM20C expression in osteoblasts and subsequent DMP1 secretion into ECM during periadolescence.

## Discussion

In this study, we identify a two-step molecular mechanism by which mechanical forces drive the maturation of bone and vessels at the end of adolescence (Fig. 7). First, body weight-associated mechanical loading triggers the mechanoreceptor PIEZO1 to enhance the production of FAM20C kinase in osteoblasts, which induces a burst in DMP1 secretion into ECM. Second, large

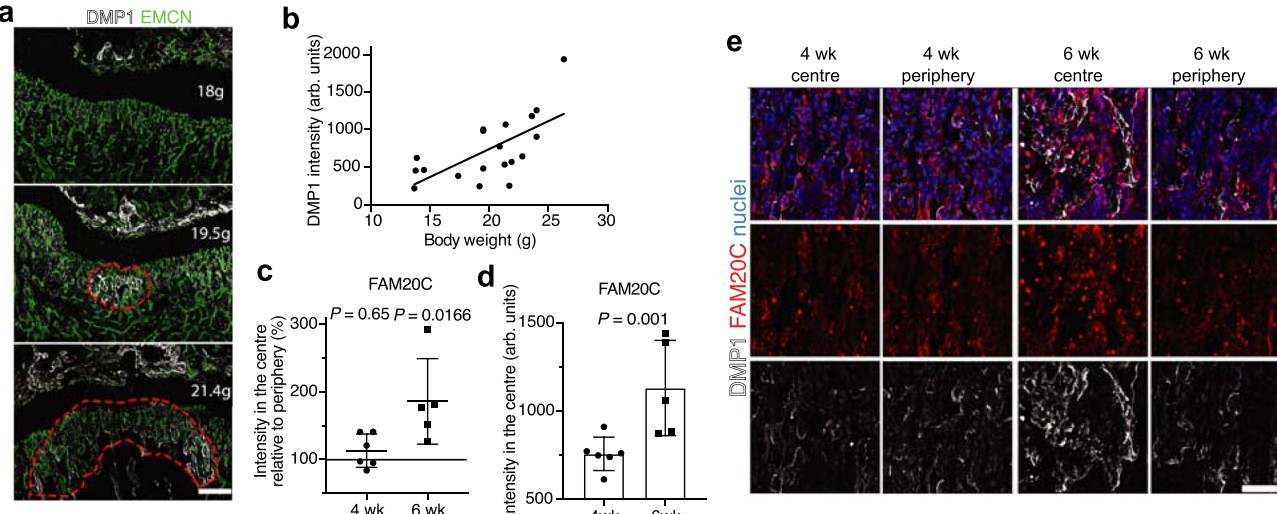

**Fig. 4 DMP1 secretion correlates positively with body weight and coincides spatiotemporally with FAM20C upregulation. a** DMP1 and EMCN immunostaining of 5.5-week-old femurs derived from littermate mice. Red dotted line marks the area of extracellular DMP1 at the OF. Scale bar 250 µm. **b** Counterplot of body weight versus DMP1 intensity. $R^2 = 0.4139$, $n = 19$. **c** Quantification of FAM20C intensity in the centre relative to periphery (centre + periphery) in 4- and 6-week-old femurs. Two-tailed paired $t$-test. **d** Quantification of FAM20C intensity in the centre in 4- and 6-week-old femurs. Two-tailed paired $t$-test. **c, d** Pooled data from two independent experiments. At 4 weeks $n = 6$, 6 weeks $n = 5$. **c, d** Source data are provided in Source Data file. **e** FAM20C and DMP1 immunostaining in OF of 4- and 6-week-old femurs. Scale bar 50 µm.

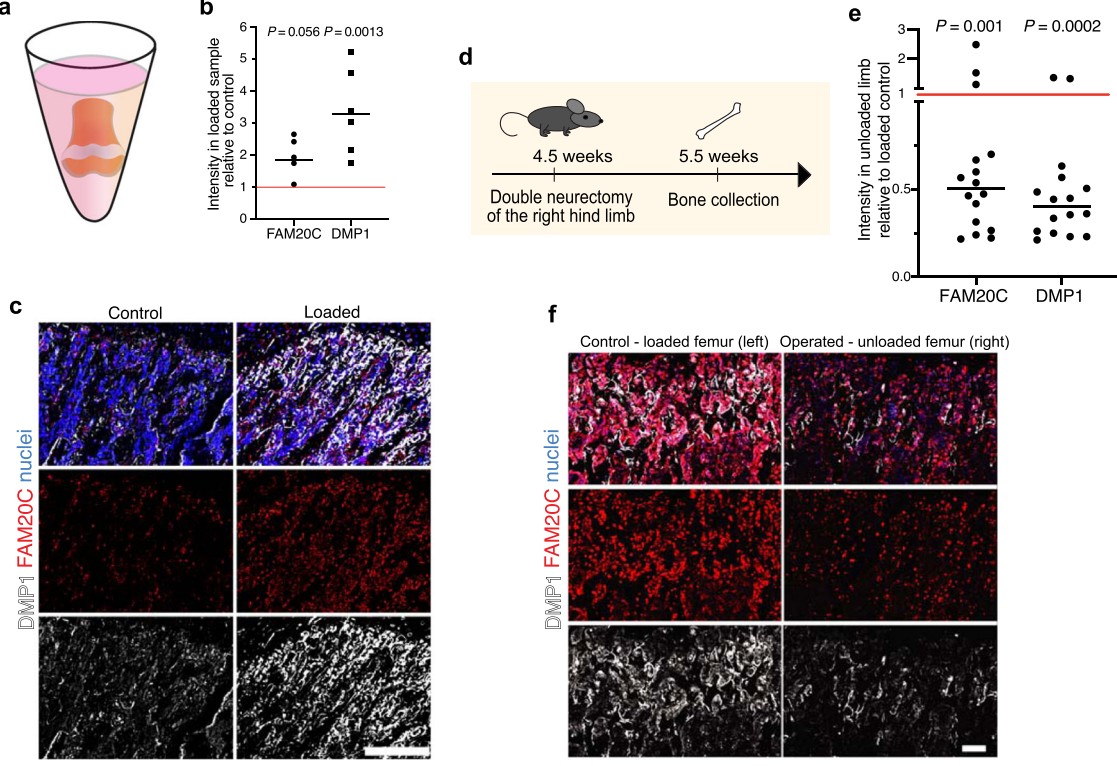

**Fig. 5 Mechanical loading enhances FAM20C kinase production and DMP1 secretion. a** Schematic depiction of the position of a femur slice (≈500 µm thick) derived from 4-week-old mice in a well of a 96-well PCR plate filled with cell culture medium. The slices were either just cultured for 5 days (control) or exposed to loading via low-speed centrifugation (30 min. at 19 g) twice per day (loaded). **b** Quantification of FAM20C and DMP1 intensities in the loaded (centrifuged) samples relative to control (non-centrifuged) group (red line). Pooled data from two independent experiments. Mean, $n = 5$–6. $P$ values for paired analysis of samples derived from one mouse. Two-tailed paired $t$-test. **c** FAM20C and DMP1 immunostaining of control and loaded (centrifuged) femur slices. Scale bar 100 µm. **d** Experimental setup for hind limb unloading via sciatic and femoral double neurectomy at the age of 4.5 weeks. **e** Quantification of FAM20C and DMP1 intensities in the unloaded femur relative to the loaded control (red line) at 5.5 weeeks. $P$ values show significance after paired analysis of intensities derived from the same mouse. Pooled data from two independent experiments. Two-tailed Wilcoxon matched-pairs signed rank test. Median, $n = 16$. **b, e** Source data are provided in Source Data file. **f** FAM20C and DMP1 immunostaining of loaded (control, left hind limb) and unloaded (operated, right hind limb) femurs at 5.5 weeeks. Scale bar 50 µm.

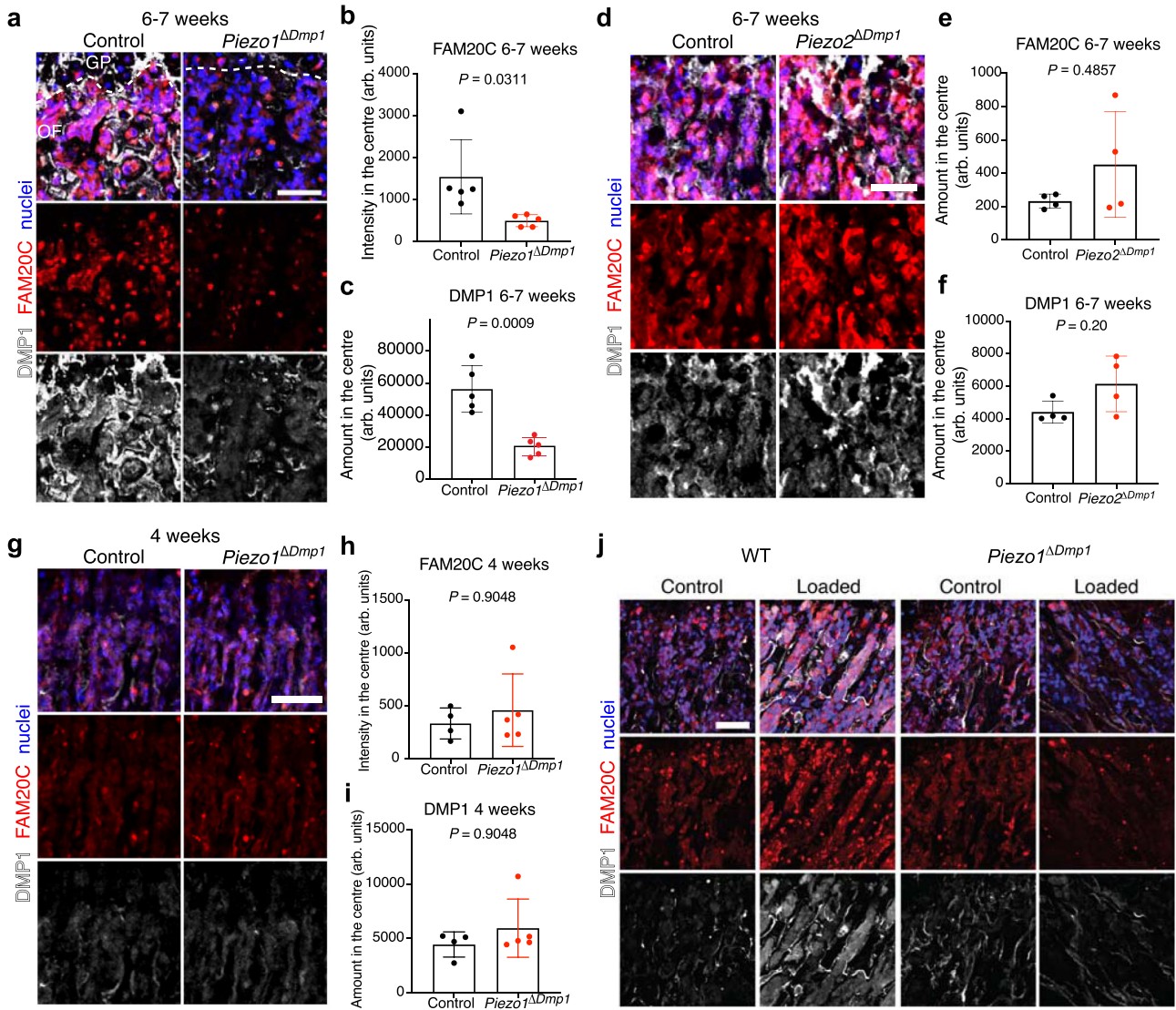

**Fig. 6 Mechanoreceptor PIEZO1 drives FAM20C upregulation in osteoblasts and DMP1 secretion during periadolescence. a** FAM20C and DMP1 immunostaining of 6-7-week-old control and Piezo1$^{\Delta Dmp1}$ OF. Scale bar 50 µm. **b**, **c** Quantification of FAM20C (**b**) and DMP1 (**c**) intensities in 6-7-week-old control and Piezo1$^{\Delta Dmp1}$ OF. n = 5. Two-tailed unpaired t-test. **d** FAM20C and DMP1 immunostaining of 6-7-week-old control and Piezo2$^{\Delta Dmp1}$ OF. Scale bar 50 µm. **e**, **f** Quantification of FAM20C (**e**) and DMP1 (**f**) intensities in 6-7-week-old control and Piezo2$^{\Delta Dmp1}$ OF. n = 4. Two-tailed unpaired t-test. **g** FAM20C and DMP1 immunostaining of 4-week-old control and Piezo1$^{\Delta Dmp1}$ OF. Scale bar 50 µm. **h**, **i** Quantification of FAM20C (**h**) and DMP1 (**i**) intensities in 4-week-old control and Piezo1$^{\Delta Dmp1}$ OF. Two-tailed unpaired t-test. Mean ± SD. **b**, **c**, **e**, **f**, **h**, **i** Source data are provided in Source Data file. **j** FAM20C and DMP1 immunostaining of control and loaded (centrifuged) femur slices derived from 4-week-old Piezo1$^{\Delta Dmp1}$ and Cre-negative (WT) littermate mice. Scale bar 50 µm.

amounts of extracellular DMP1 inhibit VEGF signalling in the OF and transform highly angiogenic type H vessels into quiescent type L vasculature to limit bone growth activity. In parallel, extracellular DMP1 leads to rapid matrix mineralisation and strengthening of long bones[9,34]. Thus, the role of DMP1 in bone is multifaceted: it combines active inhibition of angiogenesis with mineralising activity. The discovered link between mechanical loading and FAM20C upregulation, a major kinase of the bone phosphoproteome[27,28], provides a mechanistic explanation for the well-known positive effect of loading on bone mineralisation[35]. Furthermore, it may offer an additional mechanistic explanation for hormone-independent osteoporosis caused by lack of loading, for example in astronauts[36,37] or after long immobilisation of a fractured bone[38].

Growth-promoting type H vessels have received much attention in the last years because of their substantial clinical

potential[6,39]. Our finding that inducible *Dmp1* deletion led to increased vessel activity also in subchondral bone suggests that extracellular DMP1 may protect articular cartilage from vessel invasion, which accompanies neoinnervation of the tissue and sensation of pain in osteoarthritis patients[40,41]. Moreover, DMP1-mediated inhibition of angiogenesis in bone may be useful in arresting growth of tumours including osteo- and chondrosarcomas[42–44]. In future studies, we will address the precise molecular mechanism of how extracellular DMP1 inhibits VEGF signalling in type H endothelium. Particularly we will investigate if the inhibition of VEGFR3 expression is mediated directly by DMP1 or if it is an indirect effect via the inhibition of VEGFR2 signalling[21].

Under the influence of systemic hormones, bones of male mice acquire higher mass and cortical thickness than females, which is already evident at early puberty[45] (3–5 weeks). As the weight of

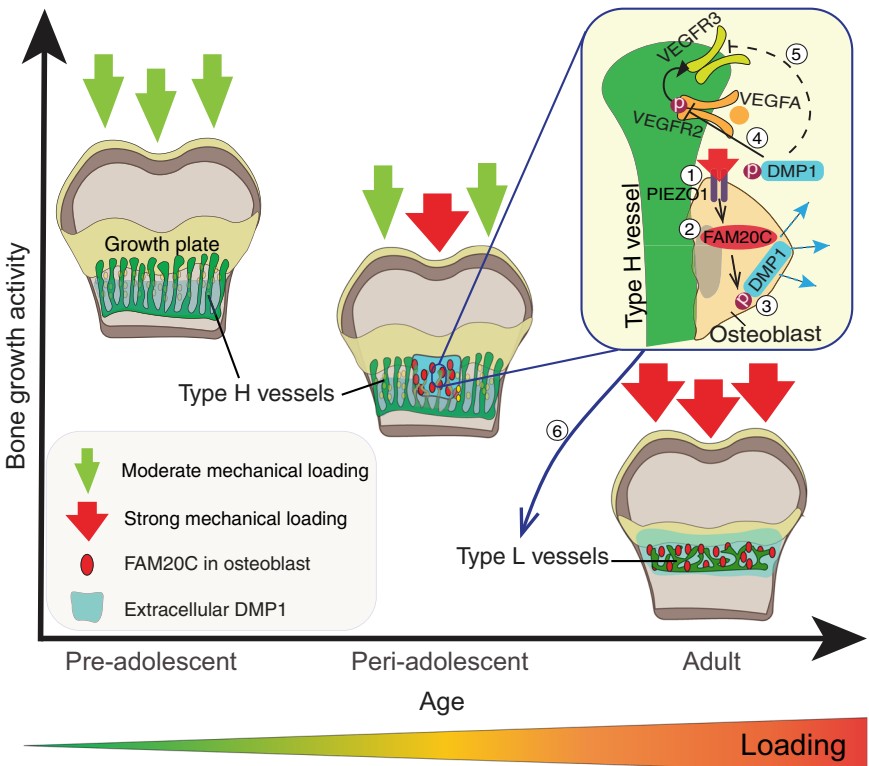

**Fig. 7 Molecular mechanism transforming type H into type L vessels to limit adolescent bone growth activity.** At the age of preadolescence, body weight-associated mechanical loading at the OF-GP border is only moderate, thus allowing the tip cells of type H vessels to invade the GP and promote bone elongation. During periadolescence, mechanical loading through increased body weight and/or muscle contractions triggers the mechanoreceptor PIEZO1 (1) to enhance the production of FAM20C kinase (2) in osteoblasts, which induces a burst in DMP1 secretion into extracellular matrix (3). Large amounts of extracellular DMP1 inhibit VEGF signalling in the OF by preventing the phosphorylation of VEGFR2 (4) and – presumably indirectly via reduced activation of VEFGR2 – the expression of VEGFR3 (5) on the tip cells of type H endothelium. Hence, highly-angiogenic type H vessels are transformed into quiescent type L vasculature to limit bone growth activity (6). In parallel, extracellular DMP1 leads to rapid matrix mineralisation and strengthening of long bones.

male mice is also greater than females at this age it is likely that the load perceived by individual osteoblasts in male and female bones is equivalent during growth according to the principles of isometric scaling[46]. This may explain why the triggering of FAM20C upregulation and DMP1 secretion happens in both genders at the same age despite the difference in absolute body weight, and presumably after a key mechanical threshold has been reached. Our results suggest that mechanical force is a key player in the control of angiogenesis in bone and hence bone growth, acting locally on bones in addition to systemic sex hormones[24–26]. However, the signalling downstream of PIEZO1 that results in enhanced FAM20C expression remains to be elucidated. Moreover, potential effects of sex hormones on PIEZO1 receptor sensitivity should be analysed in future studies.

Based on our finding that FAM20C upregulation is induced by loading and it happens first in the OF centre, we hypothesise that the central part of the OF is exposed to the highest mechanical strain since it is a spot where the loading from all four domes of the GP gets integrated. Moreover, there is no cortical bone underneath to absorb this loading (Supplementary Fig. 8r). The translation of these findings into two-legged humans, where the loading distribution and control of bone growth might be different, remains to be examined in follow-up studies.

The data presented here demonstrate that mechanical forces transmitted through PIEZO1 play a crucial role in the regulation of skeletal development and bone maturation via FAM20C-mediated extracellular DMP1 accumulation and its control of blood vessel subtypes in bone. This knowledge may be applied in bone healing studies[47], as well as in different kinds of bone

pathologies featuring abnormal bone growth activity, including osteoarthritis, osteosarcoma and osteoporosis.

## Methods

**Mice**. Tie2-GFP reporter mice (Tg(TIE2GFP)287Sato/J, Stock No: 003658) were purchased from Jackson Laboratory. For analysis of wild-type femurs, C57BL/6N and C57BL/6J mice were used. C57BL/6J mice were used for double neurectomy experiments. Dmp1$^{-/-}$ and Dmp1$^{fl/fl}$ mice were produced by microinjection of C57BL/6J zygotes with Cas9 protein and synthetic guide RNA gDMP#1 (TTGGGAAGATAACCGCTTAG) and gDMP#2 (CAGCTGAGGCGAGTAC-CACA). Thereby Exon 6, which encodes 80% of the DMP1 protein, was deleted to generate constitutive Dmp1-deficient mice (Supplementary Fig. 4). For inducible deletion of Dmp1, loxP sites were inserted at the site of the gRNA target sites flanking Exon 6 using a customised gene targeting vector: 500 bp upstream of Exon 6 and 500 bp downstream of the open reading frame (ORF). Dmp1-floxed mice were bred with Col1a2-CreERT2 mice, which were kindly provided by Prof. Dr. Thomas Blankenstein (Charité, Berlin). To induce Cre activity and gene deletion, 4-week-old offspring (both CreERT2$^{+/-}$ and also CreERT2$^{-/-}$ controls) was injected intraperitoneally with 50 mg/kg of tamoxifen (Sigma, T5648) for 5 consecutive days: P28-32. The resulting Dmp1$^{iΔOB}$ (CreERT2$^{+/-}$Dmp1$^{fl/fl}$) mutants and control (CreERT2$^{-/-}$Dmp1$^{fl/fl}$) mice were killed on day P42 (6 weeks) and the bones were collected for analysis. Piezo1$^{fl/fl}$-Dmp1Cre and Piezo2$^{fl/fl}$-Dmp1Cre were described before[28]. Cre-negative littermates were always used as controls. The vasculature phenotype of tamoxifen-injected CreERT2$^{-/-}$ control mice was identical to regular WT mice (cf. Fig. 2a (Centre) to Fig. 2h (Control)). Thus, injection of tamoxifen as such did not affect the vasculature at the doses administered.

In all experiments, both genders were analysed together and no gender-specific differences were observed. All mice were bred under specific pathogen-free (SPF) conditions and treated according to the requirements of the German and UK legislation. Mice were kept in approved animal-care facilities and were housed 4–6 per cage in standard individually ventilated cages, maintained with a 12 h light/dark cycle at an ambient temperature of 21 °C and 45% to 65% relative humidity with wood shavings bedding and nesting material. Mice had ad libitum access to tap water and standard rodent chow (Altromin - LASQCdiet™ Rod16, Autoclavable, 10 mm).

The age of mice, when they were used for the experiment was between 3 and 15 weeks, the precise age is always indicated in the text. All animal experiments were performed according to relevant laws and institutional guidelines (DRFZ, Berlin, Germany and University of Oxford, Oxford, UK) and were approved by German and UK animal ethics committees (LAGeSo license number G0175/19 and Project (Vincent) license as granted by the UK Home Office).

**Laser microdissection**. *Tie2*-GFP mice were killed via cervical dislocation at 4 weeks or 12 weeks of postnatal development. Femurs were immediately collected and frozen in SCEM medium using dry ice-isopropanol-hexane cooling bath. After freezing the blocks were kept at −80 °C until cryosectioning. The temperature of the cryostat chamber and the knife was −32 °C and -34 °C, respectively. For cryosectioning, type D knife for hard tissue was used. All surfaces including the knife were cleaned with RNaseZap before use. Cryosectioning was performed with Kawamoto's films for LMD. To minimise RNA degradation, 12 μm sections were mounted into the metal frame directly after cutting and transported in the eva-porating phase of liquid nitrogen. Laser microdissection was performed using Leica LMD7 device with fluorescent lamp for endothelial cell identification. To minimize the humidity during microdissection, constant flow of evaporating liquid nitrogen was applied. The microdissected pieces were collected into lysis buffer (ARC-TURUS® PicoPure® RNA Isolation Kit). Approximately 50 pieces were collected from each sample. Immediately after collection, the Eppendorf tube containing the sample was briefly centrifuged and incubated at 42 °C for 10 min. Afterwards, it was vortex shortly and frozen on dry ice and kept at −80 °C until RNA isolation.

**RNA sequencing and analysis**. RNA was isolated using ARCTURUS® PicoPure® RNA Isolation Kit according to manufacturer's instructions. Illumina libraries were generated using SMART-Seq v4 Ultra Low Input RNA Kit (Takara Clontech) and Nextera XT DNA Sample Preparation Kit (Illumina), with up to 10 ng of purified cDNA. The quality of synthesised cDNA was checked using Bioanalyzer (Agilent). For sequencing, Illumina NextSeq500 device was used generating 75 bp paired-end reads. Reads were mapped to the mouse mm10 genome using TopHat2 (PMID: 23618408) and Bowtie2 (PMID: 22388286) with default settings[48]. Further analysis was done with R 3.4.0 using the default settings of the deseq2 package (PMID:25516281). To identify the genes with the highest significance, fold change and decent expression level, the following filtering strategy was applied: mean expression value >500, $Log_2$ FC > 0.66, $P < 10^{-5}$. The heat map for selected genes was generated by plotting log2-transformed read counts.

**Immunohistochemistry**. Mouse bones were processed as described previously[49]. Briefly, the bones were collected, cleaned from surrounding tissue and muscles, and immediately fixed in 4% freshly prepared ice-cold paraformaldehyde (PFA) for 4 h on ice. Afterwards, the bones were washed 3 times in ice-cold PBS followed by decalcification in 0.5 M EDTA solution (pH=7.5) for 24 h under constant agitation at 4 °C. Cryoprotection was done overnight in 20% sucrose and 2% PVP solution at 4 °C immediately after decalcification. Afterwards, 5 ml of embedding medium (8% gelatine, 20% sucrose and 2% PVP in PBS) was added per falcon tube and the bones were incubated in it for 45 min at 60 °C followed by embedding into cryo-molds. The femurs were always oriented the same way with the anterior side facing the bottom of the mould. After embedding the samples were left at room tem-perature (RT) for 30 min for complete medium solidification. The blocks were kept at −80 °C in air-tight containers until cryosectioning. Cryosectioning was per-formed using NX70 ThermoFischer cryostat at −25 °C and −27 °C for chamber and blade, respectively. The disposable low-profile blades N35 for hard tissue (207500006, Feather) were used for cutting. 100 μm sections were generated and transferred onto Superfrost slides (J1800AMNT, Thermo Fisher Scientific). The slides were kept at −20 °C until staining. For immunostaining, bone sections were air-dried, permeabilized in 0.3% Triton X-100 for 15 min, blocked in 10% donkey serum at RT for 30 min, and incubated with the primary antibodies diluted in 5% donkey serum in PBS for 2 h at RT or overnight at 4 °C. After primary antibody incubation, sections were washed with PBS three times and incubated with cor-responding secondary antibodies for 1 h at RT. Nuclei were counterstained with Hoechst (1:1000, 33342, Thermo Fischer Scientific). After incubation with sec-ondary antibodies, sections were washed three times with PBS and mounted with Fluoromount-G medium (00-4958-02, Thermo Fischer Scientific). The list of used primary and secondary antibodies is provided in the Supplementary Information (Supplementary Table 1 and 2).

**RNAScope**. Freshly dissected mouse femurs were fixed in 4% freshly prepared PFA for 24 h at 4 °C followed by washings in PBS. Cryoprotection was performed via overnight incubations in 10, 20, and 30% sucrose at 4 °C. Bones were frozen in SCEM medium. The frozen blocks were kept at −80 °C until cryosectioning. Kawamoto cryofilms type 3C(16UF) were used for cryosectioning. The sections were air-dried in the cryotome for 15 min and stored at −80 °C in the slide box with molecular sieves inside. For RNAScope assay, sections were air-dried at RT for 5 min, washed in PBS for 5 min and baked on 60 °C hot plate for 30 min followed by additional fixation in 4% PFA at 4 °C for 15 min. Afterwards, the sections underwent 5 min dehydration steps in ethanol gradient: 50, 70, and 100% (x2). The sections were air-dried again for 5 min followed by 10 min incubation in hydrogen

peroxide and washings in distilled water. The target retrieval buffer was prewarmed in the steamer for ~10 min up to 75 °C. The slides were submerged into it and left in the steamer for 10 min. The temperature in the steamer was 95 °C. Afterwards, the slides were washed in distilled water and dehydrated in 100% ethanol for 10–15 s. The slides were air-dried for 5 min at RT or overnight. Protease III was applied, and the slides were incubated in the Hybez Oven for 30 min followed by washings in distilled water. The rest of the RNAScope procedure was performed according to the standard RNAScope® Multiplex Fluorescent v2 Assay protocol.

**Double neurectomy**. Mice received pre-emptive pre-operative subcutaneous analgesia. In all, 5 mm pieces of sciatic and femoral nerves of the right hind limb were cut out in 4-4.5-week-old C57BL/6J mice under general anaesthesia using isoflurane. After surgery, the mice were housed without tubes and on soft bedding. It was observed that all mice dragged the operated right limb without stepping on it. Mice were killed at the age of 5.5–6 weeks and the bones were collected for further analysis. The contralateral left femur was always used as control and was analysed in pair with the operated unloaded right femur.

**In vitro culture of bone slices**. Femurs were collected from 4-week-old C57BL/6 J mice. Half of the bone together with knee metaphysis was cut and frozen in 2.5% agarose on ice with subsequent cooling in liquid nitrogen. Under the sterile work bench, the frozen block was fixed on the sample holder for cryosectioning, which was placed on dry ice. 400−500 μm-thick slices of femur were produced using sterile disposable blade for hard tissue. Every slice was placed into a well of a 96-well PRC plate filled with StemMACS™ OsteoDiff Media (Miltenyi Biotec). The position of the slice is depicted in Fig. 4a. At least two slices per femur were generated, each of them was placed into a separate plate. The plates were kept at 37 °C with 5% $CO_2$ in a humidified atmosphere. One plate with slices was cen-trifuged twice daily at 19 g (the lowest possible speed of the centrifuge) at 37 °C for 30 min. The control plate was not subjected to centrifugation. The medium was changed every 2.5 days. After 5 days of culture, the slices were fixed in 4% PFA on ice for 30 min followed by washings in PBS and decalcification for 30 min in 0.5 M EDTA on the cooling element under constant agitation. Cryoprotection was done in 20% sucrose and 2% PVP solution in PBS at 4 °C overnight. The slices were frozen in embedding medium (8% gelatine, 20% sucrose and 2% PVP in PBS). 100 μm sections were prepared using the NX70 Cryotome (Thermo Fischer Sci-entific). The sections were stored at −20 °C until immunostaining. The staining was performed as described in the paragraph "Immunohistochemistry". Paired analysis was done based on the slices derived from the same mouse.

**Image acquisition**. All immunofluorescent and RNAScope-multifluorescent stainings were acquired at high resolution with Zeiss LSM-880 confocal microscope using 20x or 63x objectives. All microscope settings including laser power were kept constant throughout the experiments within the same type of samples and same staining panels. All immunostained samples were always acquired in Z-stack mode followed by maximum intensity projection.

**Image analysis**. The images were processed and analysed using ImageJ and CellProfiler software, in compliance with *Nature's* guide for digital images. Quantification of immunofluorescent images: first, the total intensity of a fluor-escent staining in a region of 300 μm width and 150 μm depth (from the growth plate towards the caudal region) was measured. Next, the area occupied by the protein was calculated by creating a mask (cf. Supplementary Fig. 2). Then the normalised intensity was calculated by dividing the total intensity by the occupied area (DMP1, EMCN) or by the total cell density (FAM20C). When the total amount of a protein per region was analysed, we used the total intensity without further normalisation to the area occupied by it (DMP1, pVEGFR2, MMP9). Background subtraction using ImageJ software was always done prior to quantification.

For RNAScope-multifluorescent images, the central and peripheral regions of 600 μm × 300 μm area underneath the growth plate were analysed. A cell was counted as *Runx2*+ or *Osx*+ if it contained at least one bright dot of the corresponding probe at 200x magnification. The final figures were compiled using Adobe Illustrator 2021 software (version 25.0.1).

**Endothelial cell culture**. The mouse capillary endothelial cell line bEnd.3[50] was kindly provided by Prof. Dr. Alf Hamann (Charité, Berlin). The cells were cultured in ATCC-formulated Dulbecco's Modified Eagle's Medium (Catalogue No. 30-2002) containing 10% FCS and 100 U/mL penicillin and 100 mg/mL streptomycin. The cells were maintained at 37 °C with 5% $CO_2$ in a humidified atmosphere. Cells were cultured with recombinant DMP1 (40 nM) for 4 or 24 h. At the end of this incubation time, VEGF (50 ng/ml) was added to the cells for 15-20 min with subsequent cell fixation for immunocytochemistry or RNA collection for qPCR.

**Immunocytochemistry**. bEnd.3 cells were cultured in 8- or 16-well slide chambers (Tissue Tek). By the end of the experiment, cells were fixed in 4% freshly prepared ice-cold PFA for 10 min. Afterwards, they were washed 3 times with PBS and

blocked in PBS containing 10% FCS for 30 min. The primary antibodies were applied for 1 h at RT in PBS containing 5% FCS and 0.1% Tween-20. After primary antibody incubation, cells were washed 3 times with PBS-Tween-20 (0.1%) and incubated with appropriate secondary antibodies diluted in PBS containing Hoechst (1:1000) for 1 h at RT. Finally, the cells were washed again three times with PBS and mounted using Fluoromount-G medium (Thermo Fischer Scientific).

**qPCR**. mRNA of bEnd.3 cells was isolated with the NucleoSpin RNA XS kit (Macherey Nagel) according to the manufacturer's protocol. cDNA was generated with TaqMan Reverse Transcription Reagents and quantified with PowerUp™ SYBR™ Green Master Mix using QuantStudio Real-Time PCR System (Thermo Fischer Scientific). The following primers were used: *Fos* (forward: CACTC-CAAGCGGAGACAGAT, reverse: TCGGTGGGCTGCCAAAATAA), *Gapdh* (forward: AAGGTGATCCCAGAGCTGAA, reverse: CTGCTTCAC-CACCTTCTTGA). *Fos* expression was normalised to *Gapdh*.

**Von Kossa staining**. The samples were prepared as described in paragraph "RNAScope". The staining was done according to the manufacturer's instructions (Silver plating kit acc. to von Kossa, Merck Millipore, #100362). Briefly, 10 μm undecalcified femur sections were placed in distilled water for 1 min followed by incubation in silver nitrate solution for 5 min under direct sunlight. Afterwards, the samples were washed in running tap water for 3 min and incubated in sodium thiosulfate solution for 5 min. Then the samples were washed in running tap water for 1 min and mounted in water-based mounting medium. The images were acquired with Leica DM750 equipped with Leica ICC50 W Camera Module using LAS software and analysed with ImageJ software (version 2.0.0).

**Histomorphometry**. Dissected skeletons were fixed in 3.7% PBS-buffered formaldehyde for 18 hours, before they were stored in 80% ethanol. Bones were dehydrated in ascending alcohol concentrations and then embedded in methyl-methacrylate for undecalcified histology as described[51]. Histological sections of 4 μm thickness from the sagittal plane were stained by von Kossa/van Gieson procedures as described[52]. Histomorphometry was performed according to the ASBMR guidelines using the OsteoMeasure histomorphometry system (Osteo-Metrics, Decatur, GA, USA).

**Statistics**. Statistical analysis was carried out using GraphPad Prism software (version 8.0d and 8.4). Normal distribution was tested using Shapiro-Wilk and Kolmogorov–Smirnov tests. In case of normal distribution parametric two-tailed paired *t*-test or unpaired *t*-test were used. For non-Gaussian distribution, non-parametric tests were used: two-tailed Wilcoxon test for paired analysis and two-tailed Mann–Whitney test for unpaired analysis. $P < 0.05$ was considered significant. Mice with body weight that was >20% below the average weight of their littermates were excluded from the inducible *Dmp1*-deletion experiments. All image quantifications were done automatically upon manual blinded selection of central and peripheral OF regions of the same size. Sample numbers are indicated in figure legends. Reproducibility was ensured by several independent experiments.

**Reporting summary**. Further information on research design is available in the Nature Research Reporting Summary linked to this article.

## Data availability

RNA sequencing data are available at gene expression omnibus (GEO) under the accession number GSE148804. Source data are provided with this paper. All other data supporting the key findings of this study are available from the corresponding authors upon request.

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

## Acknowledgements

We thank Prof. Dr. T. Blankenstein for providing *Col1-CreERT2* mice; Prof. Dr. A. Hamann for providing bEnd.3 cell line; Dr. G. Heinz and K. Lehmann for help with cDNA library preparation; V. Holecska, I. Panse and C. Rüster for technical assistance; M. Drabkina for advice on RNAScope technique; and Dr. P. Saikali, Dr. P. Shen, N. Duran, V. Plajer, and Dr. C. Helmstetter for advice and consultation. This work was supported by the Willy Robert Pitzer Foundation (Osteoarthritis Research Program, to M.L.), the state of Berlin and the "European Regional Development Fund" (ERDF 2014–2020, EFRE 1.8/11, to M.F.M.), the Leibniz Association (Leibniz Collaborative Excellence, TargArt to M.F.M.), Deutsche For-schungsgemeinschaft (grant no. AM 103/31-1, to T.S.), Centre for OA Pathogenesis Versus Arthritis (grant no. 21621, to T.L.V.). M.D. and T.M.B. were fellows of the International Max Planck Research School for Infectious Diseases and Immunology.

## Author contributions

M.D. and M.L. designed the experiments, interpreted the results and wrote the manuscript. T.M.B. helped with the generation of the transgenic mouse lines and in vivo experiments and provided technical expertise. J.M.Z. performed and T.L.V. supervised double neurectomy experiments. F.H. and M.F.M. provided transcriptome analysis. L.B. and T.S. provided *Piezo1*$^{\Delta Dmp1}$ and *Piezo2*$^{\Delta Dmp1}$ samples. A.K., T.S., and T.L.V. provided expertise and advice. R.K. designed and generated *Dmp1*$^{-/-}$ and *Dmp1*$^{fl/fl}$ mice. M.D. performed all other experiments and analysed the data. All authors reviewed and edited the manuscript.

## Funding

## Competing interests

The authors declare no competing interests.
