## [Peer Review File · Nature Communications]

Editorial Note: Parts of this peer review file have been redacted as indicated to maintain the confidentiality of other journals.

REVIEWER COMMENTS

Reviewer #1 (Remarks to the Author):

The authors have dealt with the criticisms I brought up in the previous assessment in a satisfactory manner. I have no further comments.

Reviewer #2 (Remarks to the Author):

In this paper, the authors have shown that mechanical forces associated with increased body weight at maturity activate the mechanoreceptor PIEZO1 to stimulate the production of the kinase FAM20C in osteoblasts. The latter phosphorylates Dmp1 that inhibits VEGF signaling at the tip cells of type H endothelium. The type H vessels are thus transformed into type L vessels to limit bone growth and increase bone mineralization.

This paper was submitted to [redacted] and has been transferred to Nature Communications. In all, this is an elegant conceptual advance to further link osteogenesis with angiogenesis, and the studies particularly through the use of conditional Dmp1 deletion that separate the temporal effects from global deletion, are well done and compelling. The authors have also provided a thorough and thoughtful response to critique from three reviewers. However, several issues need to be resolved before the manuscript is acceptable.

1. While evidence for the pathway is convincing, it is not clear what type of mechanical stimulus that arises at adolescence would be different from that experienced in growing bone. The emphasis on

mechanical stimulation being the cause of the phenotypic change in the vasculature is farfetched and needs toning down. Furthermore, mechanical stimulation in two-legged humans is quite different from that of mice.

2. If the authors regard body weight as the determinant, than obese mice should show a similar effect. Do db/db mice, for example, show a vascular phenotype?

3. Adolescence is also associated with a plethora of hormonal changes—do any of these changes affect the PIEZO1 receptor. While this may be beyond the scope of this paper, the authors need to discuss this aspect as a caveat to their mechanical stimulation hypothesis.

4. Mechanistically, how DMP1 turns off VEGF signaling at the tip cells of the H vessels remains unclear. While the phenomenon is documented, how exactly DMP1 acts on the VEGFR3 remains unknown. Is it a direct action or is there an intermediary step? This needs careful discussion as a caveat, if the authors have no new data.

5. Are the authors certain that tamoxifen does not affect the vasculature at the doses administered. This needs experimental documentation, with appropriate controls.

6. There seems to be confusion between osteocytes and osteoblasts. While osteocytes are major producers of DMP1 for mineralization, osteoblasts produce considerably less DMP1. This distinction needs further clarity.

Reviewer #3 (Remarks to the Author):

I am happy with the revision the authors made and I am supporting the publication of the paper

Reviewer #2 (Remarks to the Author):

In this paper, the authors have shown that mechanical forces associated with increased body weight at maturity activate the mechanoreceptor PIEZO1 to stimulate the production of the kinase FAM20C in osteoblasts. The latter phosphorylates Dmp1 that inhibits VEGF signaling at the tip cells of type H endothelium. The type H vessels are thus transformed into type L vessels to limit bone growth and increase bone mineralization.

This paper was submitted to [redacted] and has been transferred to Nature Communications. In all, this is an elegant conceptual advance to further link osteogenesis with angiogenesis, and the studies particularly through the use of conditional Dmp1 deletion that separate the temporal effects from global deletion, are well done and compelling. The authors have also provided a thorough and thoughtful response to critique from three reviewers. However, several issues need to be resolved before the manuscript is acceptable.

1. While evidence for the pathway is convincing, it is not clear what type of mechanical stimulus that arises at adolescence would be different from that experienced in growing bone. The emphasis on mechanical stimulation being the cause of the phenotypic change in the vasculature is farfetched and needs toning down. Furthermore, mechanical stimulation in two-legged humans is quite different from that of mice.

Our reply:

We thank the Referee for this point. Indeed, we do not think that the mechanical stimulus arising at adolescence is qualitatively different from the one experienced in growing bone at younger age. Our results suggest that, associated with increased body weight and/or muscle contraction forces, a certain mechanical loading threshold is passed around periadolescence, which triggers PIEZO1 activation. We speculate that the strain in the central part of the growth plate/ossification front is greater than in the periphery as it is at this point where all 4 domes of the growth plate meet and there is no cortical bone structure underneath it to provide support. Therefore, at this place the loading/strain threshold may be passed earlier than in other locations, resulting in the local initiation of the cascade of events that lead to the enhancement of bone mineralization and the inhibition of angiogenesis. In our manuscript, we show that mechanical forces trigger PIEZO1 activation followed by FAM20C upregulation and DMP1 secretion from osteoblasts. The extracellular DMP1 is not only a key driver of bone mineralization but it also inhibits VEGF signaling in vitro and in vivo. Moreover, upon induced *Dmp1* deletion in osteoblasts, type H blood vessels invading the growth plate are maintained, indicating a central role of DMP1 in the switch from type H to type L vessels. Hence, the mechanical forces do not act directly on the vasculature but rather on osteoblasts, which then initiate the events resulting in the transformation of type H to type L vessels. Thus, we agree with the Referee that the title of our manuscript might have been too short in focusing exclusively on the switch in blood vessel subtypes. Therefore, we propose the following modification of the title to describe the sequence of events more clearly within the frame of the small number of words (~15) permitted for article titles: "Mechanical forces couple bone matrix mineralization with inhibition of angiogenesis to limit adolescent bone growth".

We agree with the Referee that the loading distribution and control of bone growth in two-legged humans might be different from the situation in mice. Hence, the translation potential of our findings remains to be addressed in future studies. Now we have added this aspect to the Discussion section (see Page 11, Lines 15-17).

2. If the authors regard body weight as the determinant, than obese mice should show a similar effect. Do db/db mice, for example, show a vascular phenotype?

Our reply:

We agree with the Referee that it would be interesting to look at the vasculature development in bones of db/db mice. However, we have to keep in mind several potential limitations of this model. First, db/db mice feature endothelial dysfunction because of diabetes (Cat et al., *Sci Rep* 2018; Bruder-Nascimento et al., *Clin Sci* 2015), which might affect the vasculature phenotype substantially in addition to the increased loading caused by obesity. Second, db/db mice turn obese already in the very first weeks of postnatal development (see the table below) and hence the skeleton can adapt to the increased loading from very early on. We consider the loading threshold as relative rather than absolute measure for every organism, meaning that if a certain organism is exposed to higher loading from the very beginning, the absolute threshold to initiate the cascade of described events will be higher than in an organism exposed to normal loading. This hypothesis is based on two facts: (1) Due to isometric scaling the loading per cell in heavy species is not significantly higher than in light ones because of the skeleton architecture that makes the force distribution more or less comparable; (2) The skeleton is able to adapt itself to the existing mechanical environment by increasing bone stiffness (mass) and thus reducing the load perceived by each osteoblast. Therefore, the higher weight of db/db mice may not necessarily result in premature bone growth cessation. Moreover, leptin is known as anabolic factor for bone formation, and leptin receptor-deficient db/db mice feature an osteoporotic bone phenotype (Williams et al., *JBMR* 2011). This leptin receptor deficiency apparently overrides the known anabolic effect of increased body weight on osteogenesis, which usually results in higher bone mass and bone mineral density (Iwaniec and Turner, *J Endocrinol* 2016; Li et al., *Elife* 2019; Burr, *JBMR* 1997). To summarize, it would be interesting to examine the vasculature in bones of db/db mice but it would be challenging to dissect the indirect effects of leptin-signaling deficiency from the direct effect of increased mechanical loading caused by obesity.

RESEARCH & FACULTY
EDUCATION & LEARNING
JAX® MICE & SERVICES
PERSONALIZED MEDICINE
NEWS
ABOUT US
GIVE

Body Weight (grams; mean ± standard deviation)						
Age (weeks)	Females			Males		
	db/db	db/+	+/+	db/db	db/+	+/+
4	17.7 ± 2.5	15.0 ± 1.4	12.9 ± 1.6	18.4 ± 2.6	17.2 ± 2.7	13.4 ± 2.0
5	24.2 ± 2.7	18.4 ± 1.5	15.6 ± 1.4	25.3 ± 3.2	21.6 ± 2.1	17.7 ± 2.0
6	30.7 ± 2.8	20.0 ± 1.7	16.7 ± 1.1	32.1 ± 2.6	24.4 ± 1.9	20.5 ± 1.7
7	35.4 ± 2.5	21.6 ± 1.6	17.9 ± 1.2	36.3 ± 2.3	26.0 ± 2.0	22.2 ± 1.5
8	38.8 ± 2.3	22.4 ± 1.2	18.8 ± 1.3	39.7 ± 2.0	27.8 ± 2.2	23.6 ± 1.4
9	41.5 ± 2.6	22.6 ± 1.7	19.2 ± 1.0	42.1 ± 2.3	28.1 ± 2.0	24.2 ± 1.3

Table showing the weight of db/db mice in comparison to db/+ and WT mice throughout postnatal development. Source: The Jackson Laboratory Website.

3. Adolescence is also associated with a plethora of hormonal changes—do any of these changes affect the PIEZO1 receptor. While this may be beyond the scope of this paper, the authors need to discuss this aspect as a caveat to their mechanical stimulation hypothesis.

Our reply:

We thank the referee for raising this important question. We added this point into the discussion section (Page 11, Lines 10-11). Our results indicate that changes in sex hormones alone are not enough to trigger the inhibition of angiogenesis and mechanical stimulation is an essential player in this process. However, we fully agree that it is unknown whether sex hormones affect the sensitivity of the PIEZO1 receptor.

4. Mechanistically, how DMP1 turns off VEGF signaling at the tip cells of the H vessels remains unclear. While the phenomenon is documented, how exactly DMP1 acts on the VEGFR3 remains unknown. Is it a direct action or is there an intermediary step? This needs careful discussion as a caveat, if the authors have no new data.

Our reply:

We fully agree that a follow-up study devoted to the precise mechanism of DMP1 action on VEGFR2 phosphorylation and VEGFR3 expression would be very interesting. According to the literature, we rather assume an indirect inhibitory effect of DMP1 on VEGFR3 expression mediated via the inhibition of VEGFR2 signaling by extracellular DMP1. However, direct inhibition of VEGFR3 expression by DMP1 is conceivable as well. Now these aspects are discussed on Page 10, Lines 26-27 of the revised manuscript.

5. Are the authors certain that tamoxifen does not affect the vasculature at the doses administered. This needs experimental documentation, with appropriate controls.

Our reply:

We shared the same concern as the Referee when we designed the corresponding experiments of our study. Therefore, we controlled the effect of tamoxifen by injecting it into all the experimental mouse groups (*CreERT2^{+/+}* and *CreERT2^{-/-}* controls). The vasculature phenotype of tamoxifen-injected *CreERT2^{-/-}* mice was identical to regular WT mice (cf. Fig. 2a (Centre) to Fig. 2h (Control)). Thus, injection of tamoxifen as such did not affect the vasculature at the doses administered. This control is now clarified in the legend of Fig. 2c and in the Materials and Methods section of the revised manuscript (Page 21, Lines 13-14 and Lines 18-20).

6. There seems to be confusion between osteocytes and osteoblasts. While osteocytes are major producers of DMP1 for mineralization, osteoblasts produce considerably less DMP1. This distinction needs further clarity.

Our reply:

To identify the main producers of *Dmp1* in the ossification front, we assessed the expression of osteoblast- and osteocyte-specific markers in parallel to the level of *Dmp1* expression throughout postnatal development using Multiplex RNAScope technology (Page 4, Lines 11-20 of the manuscript).

The main producers of *Dmp1* in the ossification front were *Runx2*- and *Osx*-positive cells (osteoblasts). We could not detect sclerostin-positive cells (osteocytes) in the ossification front at the examined stages that were central for the study (4–6 weeks of age). Instead in the cortical bone, we found many sclerostin-positive osteocytes, and these cells also expressed decent amounts of *Dmp1*. Now this information is added to the Results section (Page 4, Lines 14-15). Therefore, there is no confusion between these two cell types but rather *Dmp1* expression is not restricted exclusively to osteocytes as it also occurs in osteoblasts in the ossification front at this rather early developmental stage.

References

- Bruder-Nascimento, T., Callera, G. E., Montezano, A. C., He, Y., Antunes, T. T., Dinh Cat, A. N., Tostes, R. C., & Touyz, R. M. (2015). Vascular injury in diabetic db/db mice is ameliorated by atorvastatin: Role of Rac1/2-sensitive Nox-dependent pathways. *Clinical Science* **128**, 411–23.
- Burr, D. B. (1997). Muscle strength, bone mass, and age-related bone loss. *JBMR* **12**, 1547-51.
- Iwaniec, U. T., & Turner, R.T. (2016). Influence of body weight on bone mass, architecture and turnover. *J Endocrinol.* **230**, R115-30.
- Li, X., Han, L., Nookaew, I., Mannen, E., Silva, M. J., Almeida, M., & Xiong, J. (2019). Stimulation of Piezo1 by mechanical signals promotes bone anabolism. *ELife* **8**, e49631.
- Nguyen Dinh Cat, A., Callera, G. E., Friederich-Persson, M., Sanchez, A., Dulak-Lis, M. G., Tsiropoulou, S., Montezano, A. C., He, Y., Briones, A. M., Jaisser, F., & Touyz, R. M. (2018). Vascular dysfunction in obese diabetic db/db mice involves the interplay between aldosterone/mineralocorticoid receptor and Rho kinase signaling. *Scientific Reports* **8**, 2952.
- Williams, G. A., Callon, K. E., Watson, M., Costa, J. L., Ding, Y., Dickinson, M., Wang, Y., Naot, D., Reid, I. R., & Cornish, J. (2011). Skeletal phenotype of the leptin receptor-deficient db/db mouse. *JBMR* **26**, 1698-709.

REVIEWERS' COMMENTS

Reviewer #2 (Remarks to the Author):

I have looked at the revised manuscript in detailed. The manuscript has now taken into account all comments and is worthy of publication.